# Circuit analysis reveals a neural pathway for light avoidance in *Drosophila* larvae

Altar Sorkaç [1,2,5], Yiannis A. Savva[1,2,3,5], Doruk Savaş [1,2,5], Mustafa Talay [1,2,4] & Gilad Barnea [1,2] ✉

Understanding how neural circuits underlie behaviour is challenging even in the connectome era because it requires a combination of anatomical and functional analyses. This is exemplified in the circuit underlying the light avoidance behaviour displayed by Drosophila melanogaster larvae. While this behaviour is robust and the nervous system relatively simple, the circuit is only partially delineated with some contradictions among studies. Here, we devise trans-Tango MkII, an offshoot of the transsynaptic circuit tracing tool trans-Tango, and implement it in anatomical tracing together with functional analysis. We use neuronal inhibition to test necessity of particular neuronal types in light avoidance and selective neuronal activation to examine sufficiency in rescuing light avoidance deficiencies exhibited by photoreceptor mutants. Our studies reveal a four-order circuit for light avoidance connecting the light-detecting photoreceptors with a pair of neuroendocrine cells via two types of clock neurons. This approach can be readily expanded to studying other circuits.

Neural circuits underlie all brain functions including processing sensory information and controlling behaviour. Understanding circuit mechanisms necessitates the use of a multi-pronged approach encompassing anatomical and functional analyses. The gold standard in anatomical analysis is electron microscopy (EM) reconstruction generating a connectome, and much effort has been devoted to producing connectomes of various nervous systems of organisms with increasing complexities. However, even when studying a simple behaviour in a simple organism, analysis of several layers of connected neurons is necessary, a significant challenge when using the connectome data. Further, to truly understand the flow of information in a circuit, one must use functional approaches to manipulate elements within the circuit and observe the consequences. The light avoidance behaviour, or photophobia, exhibited by larvae of *Drosophila melanogaster* is an example of a robust behaviour in a relatively simple organism. Nevertheless, our knowledge about the neural circuit mediating photophobia is patchy, and at times contradictory[1,2]. To initiate photophobia, light is detected by Rh5 photoreceptors in the larval eye, the Bolwig Organ[1,3]. In the central brain, the prothoracicotropic hormone (PTTH)-expressing neurons are essential for photophobia[4,5]. How these two neuronal types are connected is less clear.

Here, we establish *trans*-Tango MkII, a modified version of the transsynaptic tracing tool *trans*-Tango and show that it works effectively in *Drosophila* larvae. We then use *trans*-Tango MkII to reveal the connections between Rh5 photoreceptors and PTTH neurons in the neural circuit underlying light avoidance behaviour. We corroborate our findings by circuit epistasis analysis that includes neuronal silencing to test for necessity, and activation to assess sufficiency to rescue defects in light avoidance exhibited by Rh5 null mutants. Since *trans*-Tango MkII fills a gap in larval circuit tracing, our approach constitutes a general framework for studying neural circuits in *Drosophila* larvae.

[1]Department of Neuroscience, Brown University, Providence, RI 02912, USA. [2]Robert J. and Nancy D. Carney Institute for Brain Science, Brown University, Providence, RI 02912, USA. [3]Present address: Shape Therapeutics, Inc, Seattle, WA 98109, USA. [4]Present address: Department of Molecular and Cellular Biology, Howard Hughes Medical Institute, Harvard University, Cambridge, MA 02138, USA. [5]These authors contributed equally: Altar Sorkaç, Yiannis A. Savva, Doruk Savaş. ✉e-mail: gilad_barnea@brown.edu

## Results

### Establishing *trans*-Tango MkII

The information regarding light detection by Rh5 photoreceptors could be conveyed to PTTH neurons either directly through synaptic connections or indirectly via other neurons. To reveal whether Rh5 photoreceptors are presynaptic to PTTH neurons, we decided to use *trans*-Tango, a transsynaptic circuit tracing, monitoring and manipulation tool[6,7].

While *trans*-Tango has been effectively used to reveal synaptic connections in the adult *Drosophila* nervous system, background noise in larvae limits its utility in most larval circuits[6] (Supplementary Fig 1a). We reasoned that this problem might arise from ectopic expression of the ligand in the larval ventral nerve cord (VNC). The ligand construct in *trans*-Tango comprises the intracellular and transmembrane domains of dNRXN1 to localise it to the presynaptic sites, and the extracellular domain of hICAM1 that spans the synaptic cleft to deliver the ligand (hGCG) to its cognate receptor on the postsynaptic membrane (Supplementary Fig 1b). It is, thus, conceivable that the sequences encoding hICAM1 or dNRXN1 lead to misexpression of the ligand fusion protein. To solve the misexpression problem, we screened different ligand fusion proteins that would localise to the presynaptic site. We then assayed these versions of the ligand fusion proteins in the olfactory system, initiating the system from a subset of larval olfactory receptor neurons (ORNs). Since the signal in *trans*-Tango is temperature dependent[6], we conducted these experiments at 18 °C and 25 °C. Of the presynaptic proteins we tried, full-length dNRXN1 (Supplementary Fig. 1b) yielded the best signal to noise ratio at 18 °C (Supplementary Fig. 1c). However, we could still observe reduced, albeit visible, background noise in the absence of a driver (Supplementary Fig. 1c). By contrast, when we reared the flies at 25 °C, we observed strong signal with virtually no background noise (Supplementary Fig. 1d). We termed this new version *trans*-Tango MkII. We, then, set out to characterise *trans*-Tango MkII in different circuits, in both larvae and adults.

Since the EM reconstruction of the olfactory system of the first instar larvae is available[8], we compared the *trans*-Tango MkII results in the olfactory system of the third instar larvae with the EM reconstruction data. When we initiated *trans*-Tango MkII from a subset of ORNs expressing the receptor Or42a, we counted an average of $22 \pm 3$ neurons in each side of the brain in five brains (Supplementary Fig. 1e). According to the EM reconstruction, 14 neurons (16 including single synapse connections) are postsynaptic to Or42a-expressing ORNs in one side of the brain, and 16 neurons (20 including single synapse connections) in the other side[8]. In the same brains, we counted six to seven projection neuron (PN) axons in each side (Supplementary Fig. 1f). The EM reconstruction identified five PNs as postsynaptic to Or42a-expressing ORNs in first instar larvae[8]. The fact that we see more neurons via *trans*-Tango MkII might be due to changes in the connections between first and third instar larvae. However, although some of the neurons identified by the EM reconstruction have processes in the suboesophageal zone, the density of the innervation in this area might mean that *trans*-Tango MkII exhibits some false positive signal. Nonetheless, *trans*-Tango MkII reveals the expected connections in this circuit.

Having successfully used *trans*-Tango MkII to trace connections from the periphery to the CNS, we next wished to implement it to reveal connections within the CNS. One such easily identifiable connection exists in the mushroom body calyx between the PNs and the Kenyon Cells[9,10]. To access a subset of PNs, we employed the commonly used GH146 driver that also expresses in cells outside the olfactory circuit[11–13]. When we initiated *trans*-Tango MkII from GH146-expressing PNs we observed postsynaptic signal in Kenyon cells as expected (Supplementary Fig. 1g, h). While *trans*-Tango MkII works well in the larval nervous system, we observed strong background signal when we used it in the adult brain (Supplementary Fig. 2).

To further characterise *trans*-Tango MkII, we turned to the larval visual system. EM reconstruction of the visual system of the first instar larva reveals nine strong and two or three weak postsynaptic partners for Rh5 photoreceptors in each side of the brain. For Rh6 photoreceptors, there are six strong and two or three weak postsynaptic partners[14]. When we initiated *trans*-Tango MkII from Rh5 photoreceptors of the first instar larvae, we revealed an average of ten neurons per side that were labelled as postsynaptic in three brains (Supplementary Fig. 3a). The equivalent experiment from Rh6 photoreceptors resulted in postsynaptic signal in an average of five neurons per side in four brains (Supplementary Fig. 3b). These results suggest that in the first instar stage, *trans*-Tango MkII labels approximately the same number of neurons as identified by the EM reconstruction. It is noteworthy that when we initiate *trans*-Tango MkII from Rh6 photoreceptors in first or second instar larvae, we observe signal also in non-neuronal tissue (Supplementary Fig. 3b, d). This likely originates from the activation of *trans*-Tango during embryonic development, especially since Rh5 and Rh6 are expressed in late-stage embryos[15]. We observed that more neurons were labelled when *trans*-Tango MkII signal was analysed in the second (Supplementary Fig. 3c, d) or third (Supplementary Fig. 3e, f) instar stages. These results are difficult to interpret since it is not clear whether photoreceptors in second and third instar larvae have more postsynaptic partners than in first instar larvae, or whether these are false positives. However, it is noteworthy that these experiments highlight the specificity of *trans*-Tango MkII in the third instar larvae. Initiating *trans*-Tango MkII from Rh5 photoreceptors reveal both the four pigment-dispersing factor-expressing lateral neurons (Pdf-LaNs) and the Pdf-negative lateral neuron (5th-LaN) as postsynaptic partners. By contrast, driving *trans*-Tango MkII from Rh6 photoreceptors reveals only the Pdf-LaNs but not the 5th-LaN (Supplementary Figs. 3e, f and 4a). These results are in accordance with the EM reconstruction of the visual system in the first instar larva, suggesting that these specific connections persist through the third instar stage. These experiments indicate that while the postsynaptic partners revealed by *trans*-Tango MkII may include some false positives, the technique can be reliably used in third instar larvae raised at 25 °C.

### PTTH neurons get light information via Pdf-negative clock neurons

With a version of *trans*-Tango that works well to trace circuits in larvae, we turned back to the neural circuit underlying larval photophobic behaviour. Using *trans*-Tango MkII, we found that Rh5 photoreceptors are not presynaptic to PTTH neurons (Supplementary Fig. 4b), indicating the existence of an indirect route. Which neurons connect the Rh5 photoreceptors to PTTH neurons? The pacemaker clock neurons in the larval visual system are attractive candidates since they were previously implicated in light avoidance[1]. Expression of the genes *timeless* and *period* (*per*) reveals that the larval visual system comprises nine pacemaker clock neurons: four Pdf-LaNs, the 5th-LaN, and two pairs of dorsal neurons: DN1s and DN2s (Fig. 1a)[16]. Inhibition of all clock neurons via expression of the open rectifier truncated potassium channel dORK-ΔC[1] or of the inward-rectifying potassium channel Kir2.1 (Fig. 1b) results in decreased light avoidance at 1100 or 550 lux (205 μW/cm²), respectively. This observation indicates that at least one of the clock neurons mediates the photophobic behaviour. Indeed, this functional effect is corroborated anatomically as driving *trans*-Tango MkII from the clock neurons reveals that PTTH neurons receive direct synaptic input from them (Fig. 1c).

Pdf-LaNs are viable candidates for relaying light information from the Rh5 photoreceptors to PTTH neurons since they are postsynaptic to Rh5 photoreceptors (Supplementary Fig. 4a). Further, they have been reported to directly synapse onto PTTH neurons[4]. We therefore sought to silence the Pdf-LaNs by expressing Kir2.1 under the control of two different Pdf-Gal4 drivers. Remarkably, these experiments revealed that Pdf-LaNs are not required for light avoidance behaviour

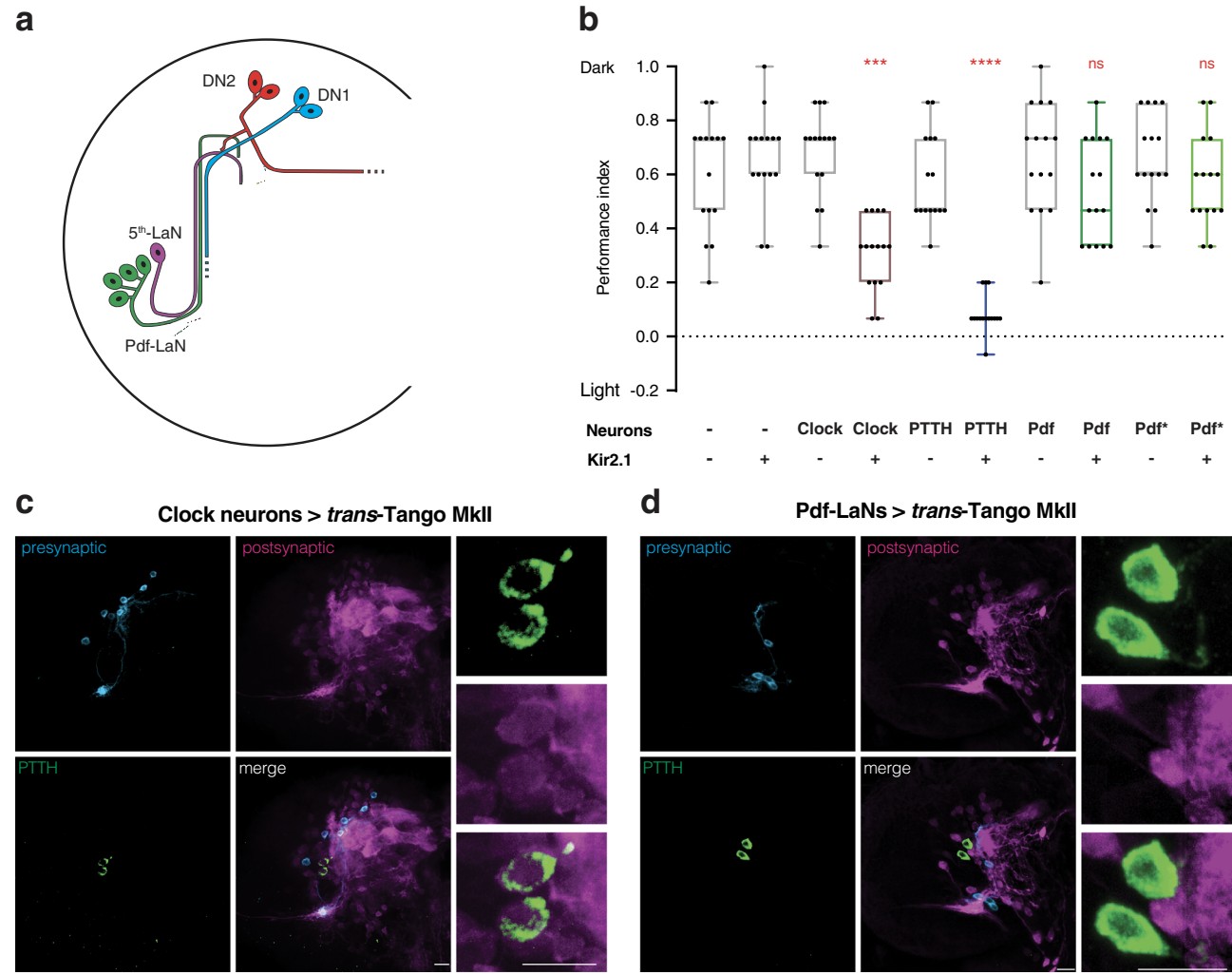

Fig. 1 | **Input from Pdf-negative clock neurons into PTTH neurons mediates light avoidance. a** Schematic of clock neurons in the *Drosophila* larval brain. **b** The effect of Kir2.1-mediated neuronal silencing on light avoidance at 550 lux. Silencing of all clock neurons or PTTH neurons decreases photophobia, silencing of Pdf-LaNs has no effect. Boxplots indicate median (middle line), 25th and 75th percentile (box), bars represent maximum and minimum. One-way ANOVA, ns: not significant,

***$p < 0.001$, ****$p < 0.0001$. $n = 15$ trials for each group. **c** Expression of the *trans*-Tango MkII ligand in all clock neurons reveals postsynaptic signal in PTTH neurons. **d** *trans*-Tango MkII reveals that PTTH neurons are not postsynaptic to Pdf-LaNs. In panels **c** and **d**, presynaptic GFP (cyan), postsynaptic mtdTomato-HA (magenta), and PTTH (green) are shown. Scale bars, 10 μm. Source data are provided as a Source Data file.

(Fig. 1b). Therefore, these cells are unlikely the link between Rh5 photoreceptors and PTTH neurons. Indeed, our *trans*-Tango MkII experiments indicate that PTTH neurons are not postsynaptic to Pdf-LaNs (Fig. 1d). This result contradicts an earlier study that reports synaptic connectivity based on GFP reconstitution across synaptic partners (GRASP) experiments. However, it is important to note that this study used a version of GRASP that is not synaptic[17], allowing for GFP reconstitution between axons in proximity. When we performed GRASP experiments using a synaptic version the technique, t-GRASP[18], we did not observe any signal for reconstituted GFP (Supplementary Fig. 5), in line with our *trans*-Tango MkII experiments.

## 5th-LaN and DN2s relay light information to PTTH neurons

To reveal which of the remaining clock neurons are presynaptic to PTTH neurons, we initiated *trans*-Tango MkII from different subsets. We genetically accessed the 5th-LaN using two drivers from the FlyLight collection[19]. In the brain, these drivers are expressed strongly in the 5th-LaN alongside weak and unreliable expression patterns in other neurons[14,20] (Supplementary Fig. 6). Initiating *trans*-Tango MkII with either driver revealed a faint postsynaptic signal in one of the PTTH neurons, suggesting a potential albeit weak connection with the

5th-LaN (Fig. 2a and Supplementary Fig. 7)[20]. We next wished to examine the two pairs of dorsal neurons. However, the drivers used to access DN1s (cry)[21] or DN2s (Clk9m)[22] also label Pdf-LaNs. Nevertheless, since Pdf-LaNs do not form synapses with PTTH neurons, any postsynaptic signal observed in these neurons would indicate direct synaptic input from DN1s or DN2s. Indeed, initiating *trans*-Tango MkII with either driver reveals that both DN1s and DN2s are presynaptic to PTTH neurons (Fig. 2b, c).

We next sought to functionally explore the role of each subset of clock neurons in photophobia using Kir2.1. In accordance with previously published results[2], we observed that silencing the 5th-LaN or DN2s leads to decreased photophobia, suggesting that these neurons are necessary for proper light avoidance. By contrast, silencing of DN1s did not affect light avoidance at 550 lux (205 μW/cm²; Fig. 3a).

We reasoned that the weak direct connection between the 5th-LaN and the PTTH neurons may not be sufficient to convey the light information from Rh5 photoreceptors. Because DN2s are presynaptic to PTTH neurons and are necessary for proper light avoidance, we initiated *trans*-Tango MkII from the 5th-LaN to examine whether DN2s constitute an indirect link. We dissected the larvae at zeitgeber time (ZT) 0 when staining with antibodies against PER reveals all clock neurons[16].

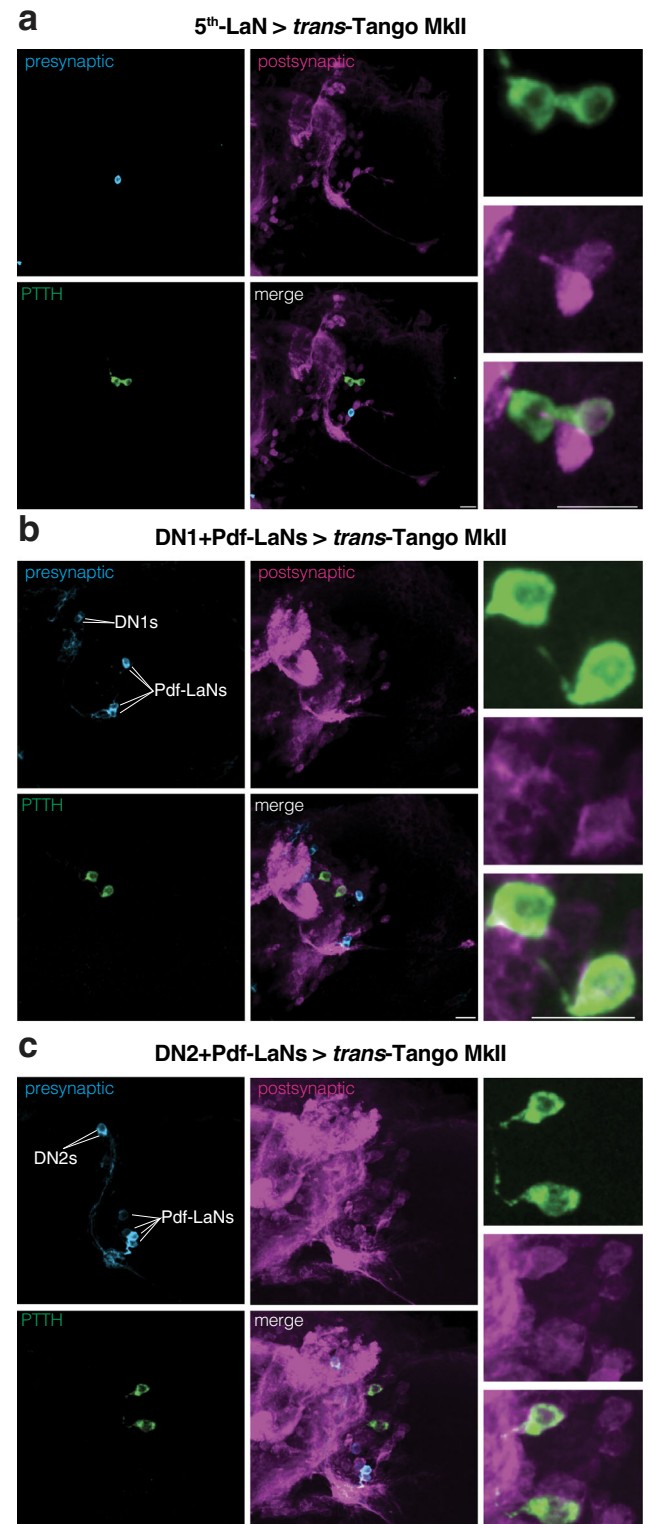

**Fig. 2 | PTTH neurons receive direct input from 5th-LaN, DN1 and DN2 clock neurons. a** Only one of the PTTH neurons receives input from the 5th-LaN. **b, c** Both PTTH neurons are postsynaptic to DN1s (**b**) and DN2s (**c**). In all panels, presynaptic GFP (cyan), postsynaptic mtdTomato-HA (magenta) and PTTH (green) are shown. Scale bars, 10 μm.

We observed that DN2s are postsynaptic to the 5th-LaN whereas DN1s are not (Fig. 3b). We confirmed these findings at ZT12 (Supplementary Fig. 8) when PER immunoreactivity is only observed in DN2s[16]. In conclusion, both our *trans*-Tango MkII and neuronal silencing experiments

revealed a possible anatomical pathway connecting Rh5 photoreceptors and the PTTH neurons via the 5th-LaN and DN2s. To further examine this flow of information, we performed functional rescue experiments by activating these subsets of neurons in Rh5 null mutants.

**Rescuing the light avoidance deficiency in Rh5 mutants**

Rh5 mutant larvae are deficient in light avoidance[1–3,23]. We reasoned that if the 5th-LaN and DN2s are indeed downstream of Rh5 photoreceptors, their activation should rescue this deficiency. To test this, we expressed the light-activated cation channel CsChrimson in different clock neurons in the Rh5 null background. CsChrimson can be excited at red wavelengths that are mostly not visible to *Drosophila*[24]. Hence, the red light used to activate CsChrimson does not, itself, cause photophobia (Supplementary Fig. 9). In the functional rescue experiments, we tested the larvae in a modified photophobia assay where half of the plate was dark, and the other half was illuminated with red light (635 nm − 451 μW/cm²) to activate CsChrimson. As we anticipated, larvae expressing CsChrimson in the 5th-LaN avoided the red-light half of the plate, whereas control animals did not exhibit any preference for either side of the plate. These results suggest that the activation of the 5th-LaN is indeed sufficient to induce aversion, and thus, to rescue the light avoidance deficiency of Rh5 null larvae (Fig. 4a). Likewise, we expected that expression of CsChrimson in DN2s would induce aversion since these neurons connect the 5th-LaN to PTTH neurons. However, to our surprise, we did not observe aversion to red light when CsChrimson was expressed using our driver for DN2s and Pdf-LaNs (Fig. 4b). Therefore, we decided to examine whether larvae expressing CsChrimson in DN1s would avoid the red-light half of the plate as DN1s are also presynaptic to PTTH neurons. We observed that these animals did not avoid the red-light part of the plate either (Fig. 4c). As is the case for DN2s, our driver for DN1s also expresses in Pdf-LaNs. Hence, we hypothesised that a potential phenotype caused by activation of the Pdf-LaNs might have masked the effects of DN1 or DN2 activation. Indeed, activation of the Pdf-LaNs alone resulted in a slight preference for the red-light half of the plate, rather than avoidance (Fig. 4d), supporting our hypothesis.

To test the effects of DN2s or DN1s exclusively, we restricted the expression of CsChrimson using the corresponding Gal4 drivers in conjunction with the Gal4-suppressor, Gal80, in Pdf-LaNs (Supplementary Fig. 10). Selective activation of DN2s was sufficient to elicit aversion (Fig. 4e), effectively rescuing the light avoidance deficiency of Rh5 null larvae. These results confirm the neural circuit that connects Rh5 photoreceptors to PTTH neurons through the 5th-LaN and DN2s.

Interestingly, selective DN1 activation also led to avoidance of the red-light half of the plate (Fig. 4f). We were puzzled by these results because DN1s are dispensable for photophobia at 550 lux (205 μW/cm²) (Fig. 3a) and their activation reduces photophobia at 750 lux[2]. Thus, DN1 activation may cause an ectopic aversion phenotype or affect a different form of photophobic behaviour. Light intensity (i.e. dim versus bright light) is an important factor in light aversion[25]. Thus, it is conceivable that DN1s mediate photophobic response to dim light. To test this possibility, we silenced DN1s at 100 lux (42 μW/cm²), and indeed observed that DN1s, but not other clock neurons, may play a role in avoidance of this light intensity (Supplementary Fig. 11). Thus, DN1s may be part of another circuit that mediates avoidance of dim light.

## Discussion

Our study revealed a circuit consisting of four orders of neurons that connect the Rh5 photoreceptors to PTTH neurons via the 5th-LaN and DN2s (Fig. 4g). While this circuit mediates the response to bright light, our observation that DN1s are necessary for photophobic response only to low light intensity indicates the existence of an additional pathway for dim light. It is noteworthy that a third, independent system has been reported in which a gustatory receptor mediates

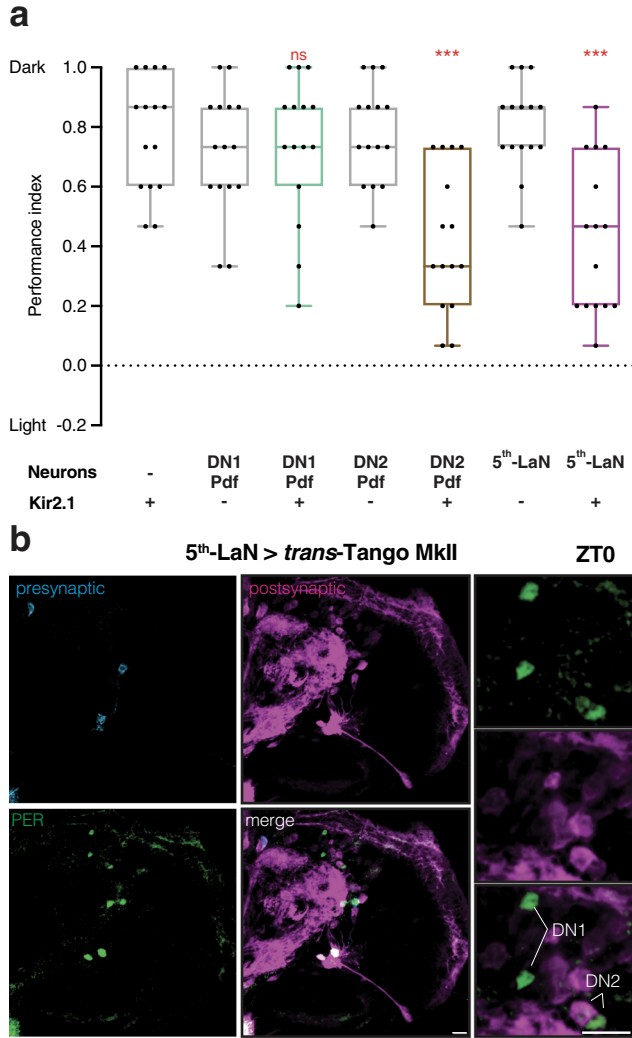

**Fig. 3 | Inhibition of either 5th-LaN or its postsynaptic partners DN2s reduces light avoidance. a** The effect of Kir2.1-mediated silencing of clock neuron subsets on light avoidance at 550 lux. Silencing of the 5th-LaN or DN2s&Pdf-LaNs results in defective photophobia, whereas silencing of DN1s&Pdf-LaNs has no effect. Box-plots indicate median (middle line), 25th and 75th percentile (box), bars represent maximum and minimum. One-way ANOVA, ns: not significant, ***$p < 0.001$. $n = 15$ trials for each group. **b** DN2s but not DN1s receive direct synaptic input from the 5th-LaN as revealed by PER staining in ZT0. Presynaptic GFP (cyan), postsynaptic mtdTomato-HA (magenta) and PER (green). Scale bars, 10 μm. Source data are provided as a Source Data file.

photophobic response to high-intensity light in class IV multidendritic neurons[26]. We do not have any information as to at which level these pathways might meet, if at all, upstream of the motor neurons.

Our results clarify several earlier studies regarding the role of Pdf-LaNs in light avoidance[1–3]. In our experiments, Pdf-LaNs are dispensable for light avoidance, yet their activation is attractive. A potential explanation is that Pdf-LaNs may modulate larval photophobia via inhibition[4,5], especially since adult Pdf-LaNs are glycinergic[27]. In addition, our results contradict a previous study reporting that Pdf-LaNs are presynaptic to PTTH neurons[4]. This study relied on a version of GRASP that is in fact not synaptic[17]. Thus, the proposed connection could have been the result of a non-synaptic reconstitution of GFP due to proximity especially since we do not observe reconstituted GFP using a synaptic version of GRASP. This result, however, does not rule out a Pdf-LaN-mediated inhibition of the light avoidance circuit from the Rh5 photoreceptors to PTTH neurons. It is conceivable that, alongside their roles in alternative circuits that

mediate this behaviour[5], Pdf-LaNs play inhibitory roles in this circuit as well. Indeed, ablating Pdf-LaNs increases the activity in PTTH neurons as revealed by the GCaMP signal[4].

Our analysis of the robust light avoidance response in larvae exemplifies the importance of employing a comprehensive approach combining circuit tracing together with neuronal inhibition and activation to test necessity and sufficiency. Our circuit epistasis analysis was made possible by *trans*-Tango MkII, a new version of *trans*-Tango that allows researchers to trace and manipulate neural circuits in *Drosophila* larvae. The combination of a robust and user-friendly genetic tool such as *trans*-Tango MkII with careful functional analysis constitutes a powerful approach that can be readily expanded to studying other circuits and behaviours.

## Methods

### Fly strains
All fly lines used in this study were maintained at 25 °C on standard cornmeal-agar-molasses media in humidity-controlled incubators under 12 h light/dark cycle, unless otherwise stated. Fly lines used in this study are in Table 1.

### Generation of transgenic fly lines
The plasmid *trans*-Tango MkII was generated using HiFi DNA Assembly (New England Biolabs #2621) and was incorporated into the attP40 locus using the ΦC31 system as described in the original *trans*-Tango paper[6]. Briefly the hICAM1::dNrxn1 sequences in the *trans*-Tango plasmid were replaced by dNrxn1 sequence amplified from the cDNA clone LP14275 (DGRC #1064347) using the following primers: 5′-atggtaacgggaatactagtCTAGATGGATCGCAAAACTCCTTCTAC-3′ and 5′-ttgttattttaaaaacgattcatggcgcgccTTACACATACCACTCCTTGACGTC-3′. The resulting PCR product was subsequently cloned via HiFi Assembly into the *trans*-Tango plasmid. All new fly strains will be deposited to Bloomington *Drosophila* Stock Center.

### Immunohistochemistry, imaging, and image processing
Larval dissections, immunohistochemical experiments, and imaging were performed as described in the original *trans*-Tango paper[6]. Unless otherwise stated, foraging third-instar larvae of either sex, or 20 day-old adult males were dissected at specified temperatures. If the temperature was not specified, the animals were reared at 25 °C. The antibodies used in this study are: anti-PDF rabbit[28] (a gift from Heinrich Dircksen, 1:3000), anti-PTTH guinea pig[29] (a gift from Michael O'Connor, 1:400), anti-PER mouse[30] (a gift from James Jepson, 1:50,000), anti-GFP rabbit (Thermo Fisher Scientific, A11122; 1:1000), anti-HA rat (Roche, 11867423001; 1:100), anti-Brp mouse (nc82; DSHB; 1:50), donkey anti-rabbit Alexa Fluor 488 (Thermo Fisher Scientific, A-21206, 1:1000), goat anti-rat Alexa Fluor 555 (Invitrogen, A21434, 1:1000), donkey anti-mouse Alexa Fluor 647 (Thermo Fisher Scientific, A-31571, 1:1000). Since the *trans*-Tango signal was too weak with R54D11-Gal4 at 25 °C, those crosses were set at 18 °C for optimisation (Figs. 2a, 3b and S4). Resultant images from *trans*-Tango experiments were processed using the Zen software (Zeiss, version 2.1) setting white, black and light corrections in all channels to provide better contrast. In *trans*-Tango figures zoomed-out images represent the maximum projection of the Z-stacks throughout the brains whereas the zoomed-in images were formed using subsets of the Z-stacks for clarity.

### Light avoidance behavioural assay
All animals used in the light avoidance behavioural assay have been 6X backcrossed to BDSC_5905. Briefly, foraging early third instar larvae of either sex were collected from the food, washed with phosphate buffered saline (PBS) twice and let dry on a surface for 3 min. In all, 13 to 16 animals were then transferred along the midline between dark and light halves of a 10-cm round petri dish with 15 mL 1.5% agar solution. Half of the lid was covered with a black tape to

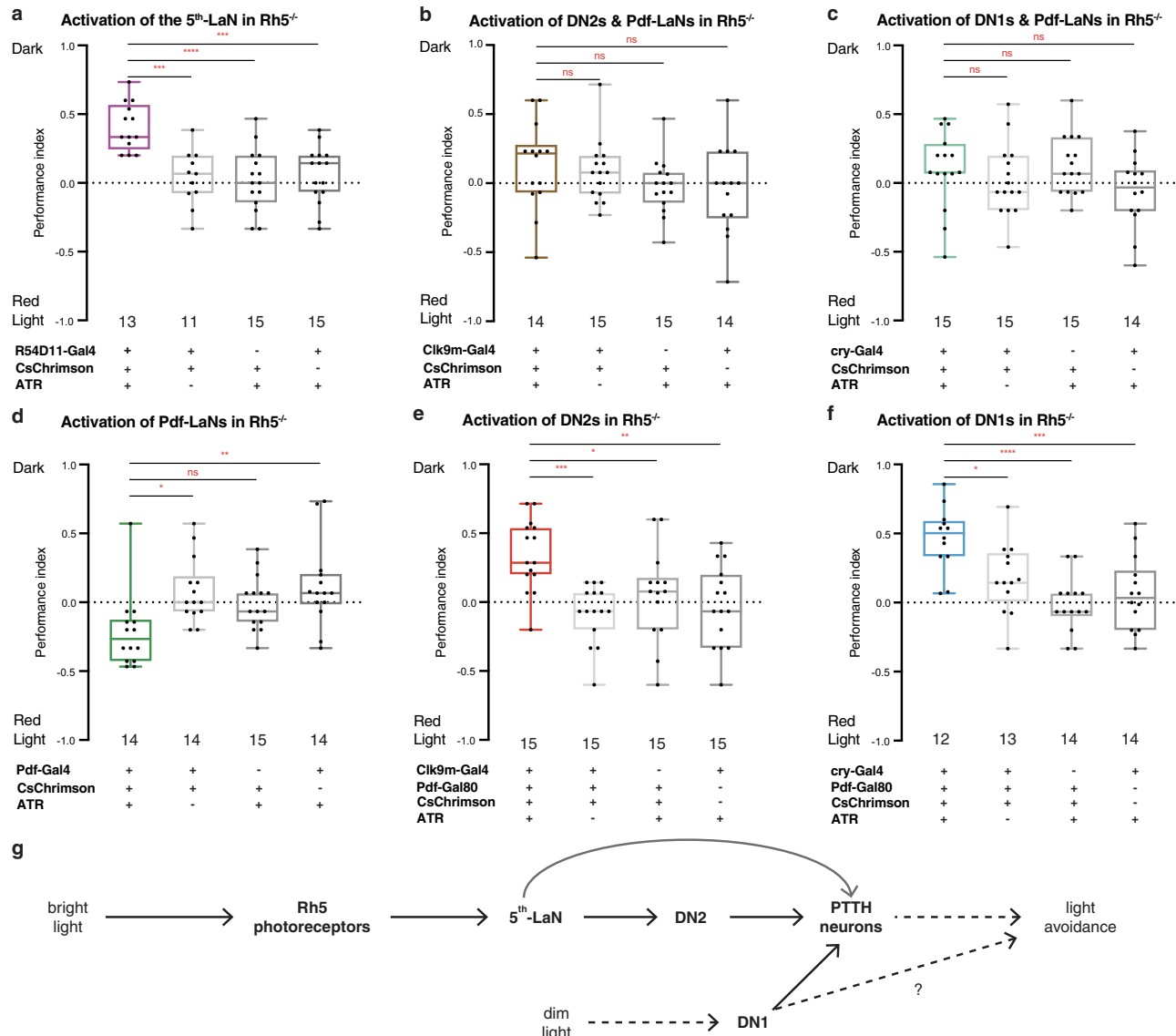

**Fig. 4 | Activation of the 5th-LaN, DN2s or DN1s rescues the light avoidance defect exhibited by Rh5 mutant larvae. a–f** Rescue of the light avoidance defect of Rh5 mutant larvae via CsChrimson mediated activation of specific subsets of clock neurons. Activation of the 5th-LaN (**a**), DN2s (**e**) or DN1s (**f**) results in light avoidance. Activation of Pdf-LaNs results in light preference (**d**). No effect is observed when Pdf-LaNs are activated alongside DN2s (**b**) or DN1s (**c**). ATR: all-*trans* retinal. Boxplots indicate median (middle line), 25th and 75th percentile (box), bars represent maximum and minimum. One-way ANOVA, ns: not significant, *$p < 0.05$,

**$p < 0.01$, ***$p < 0.001$, ****$p < 0.0001$. The number of trials for each group is indicated below each box. **g** A schematic showing the flow of information in the neural circuit that mediates the light avoidance behaviour. Bright light activates Rh5 photoreceptors that convey this information to PTTH-neurons via the 5th-LaN and DN2s to mediate light avoidance. Dim light indirectly activates DN1s that convey this information to PTTH neurons to mediate light avoidance. Source data are provided as a Source Data file.

form the dark half. The plates were exposed to 100 or 550 lux (42 µW/cm² or 205 µW/cm², respectively, measured with UDT instruments model s471 optometer, Sensor 221, aperture 1 cm²) of white light from above and experiments were run for ten minutes at 25 °C. At the end of the ten minutes, larvae on either half of the plate were counted and the preference index was calculated as (# of larvae in the dark)-(# of larvae in the light)/(total # of larvae). For each genotype/condition, at least twelve trials were run over a three-day period.

**Optogenetic rescue experiments**
Optogenetic rescue experiments were run in a similar manner to light avoidance assays with necessary modifications to accommodate for optogenetics. Instead of white light, the light half of the plates were exposed to a 635 nm LED red-light with an intensity of 1600 lux (451 µW/cm² measured with OPHIR PD200, aperture 1 cm²).

In addition, parental crosses to obtain experimental animals were set up on standard medium supplemented with 400 µM all-*trans*-retinal (ATR, Sigma #R2500) food or on a standard medium mixed with 100% ethanol for no ATR controls. All animals were kept in 24 h dark.

**Statistics and reproducibility**
For all immunochemistry experiments a representative image out of five independent brains is shown in figure panels.

The experimenter was blinded to all genotypes in the light avoidance behavioural assays. For the rescue experiments no blinding could be accommodated due to the presence of visible chromosomal balancer phenotypes. No statistical method was used to predetermine the sample size. No data were excluded from the analyses. Individual animals for each genotype were chosen randomly out of a group of larvae.

**Table 1 | Fly lines used in this study**

| *Drosophila* strains | Expression pattern | Associated figures | Stock |
|---|---|---|---|
| Clk856-Gal4[31] | All Clock neurons | 1b, 1c | |
| t-GRASP[18] | N/A | S5 | BDSC_79039 |
| Pdf-LexA[4] | Pdf-LaNs | S5 | |
| NP0394-Gal4 | PTTH neurons | 1b, S5 | DGRC_K_103604 |
| Pdf-Gal4 | Pdf-LaNs (as Pdf) | 1b, 1d, 4d | Isolated from BDSC_25031 |
| DvPdf-Gal4[32] | Pdf-LaNs (as Pdf*) | 1b | |
| R54D11-Gal4[19] | 5th-LaN, VNC, weak and unreliable expression in other neurons | 2a, 3a, 3b, 4a, S6, S8, S11a | BDSC_41279 |
| cry-Gal4[21] | DN1s and Pdf-LaNs | 2b, 3a, 4c, 4f, S10a, S11 | |
| Clk9m-Gal4[16] | DN2s and Pdf-LaNs | 2c, 3a, 4b, 4e, S10b, S11a | BDSC_41810 |
| Rh5-Gal4 | Rh5 photoreceptors | S3a, S3c, S3e | BDSC_7458 |
| Rh6-Gal4 | Rh6 photoreceptors | S3b, S3d, S3f | BDSC_7464 |
| R19C05-Gal4[14,19] | 5th-LaN, others | S7 | BDSC_48842 |
| Or42a-Gal4[33] | Or42a ORNs | S1a, S1c, S1d, S1e | BDSC_9960 |
| Or42b-Gal4[33] | Or42b ORNs | S2a, S2b | BDSC_9971 |
| GH146-Gal4[11] | Subset of PNs, others | S1f, S1g, S1d | BDSC_30026 |
| Pdf-Gal80[34] | N/A | 4e, 4f, S10a, S10b, S11b | Isolated from BDSC_80940 |
| w1118 (5905) | N/A | 1b, S9 | BDSC_5905 |
| Rh5[2][35] | N/A | 4a, 4b, 4c, 4d, 4e, 4f, S9, S10 | |
| UAS-Kir2.1[36] | N/A | 1b, 3a, S11 | BDSC_6595 |
| UAS-Cshrimson-mVenus | N/A | 4a, 4b, 4c, 4d, 4e, 4f, S10a, S10b | BDSC_55136 |
| UAS-myrGFP, QUAS-mtdTomato-HA[37] | N/A | 1c, 1d, 2a, 2b, 2c, 3b, S1c, S1d, S1e, S1f, S1g, S2b, S3, S4, S6, S7, S8 | BDSC_30118 |
| *trans*-Tango[6] | N/A | S1a, S2a | BDSC_77123 |
| *trans*-Tango MkII | N/A | 1c, 1d, 2a, 2b, 2c, 3b, S1c, S1d, S1e, S1f, S1g, S2b, S3, S4, S7, S8 | This study |

Analysis and determination of significance in behavioural assays was performed using One-way ANOVA. Experimental groups were compared to all control groups to determine significance, the lowest pairwise significance is indicated on the figures. For analysis of Supplementary Fig. 9, one sample t-test was used. For statistical analyses, Prism 9 (GraphPad) was used.

## Reporting summary
Further information on research design is available in the Nature Research Reporting Summary linked to this article.

## Data availability
All data are available in the main text or the supplementary materials. Source data are provided as a Source Data file. Raw data is available within two weeks upon request from the corresponding author. Source data are provided with this paper.

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

## Acknowledgements
We would like to thank Alex Fleischmann, Stavros Lomvardas and members of the Barnea Lab for critical reading of the manuscript. We are grateful to Heinrich Dirckson, James Jepson and Michael O'Connor for sharing reagents. This work was supported by National Institutes of Health grants R01MH105368, R21DC014333 and R01DC017146 (GB), Carney Institute for Brain Science, Suna Kıraç Fund for Brain Science (DS). Stocks obtained from the Bloomington *Drosophila* Stock Center (NIH P40OD018537) were used in this study.

## Author contributions
A.S., Y.A.S., D.S. and G.B. conceptualised the study. A.S., Y.A.S., D.S., M.T. and G.B. devised the methodology. A.S., Y.A.S. and D.S. contributed to the investigation and visualisation of the results. A.S, Y.A.S., D.S. and G.B. administered the project. A.S., Y.A.S., D.S., M.T. and G.B. wrote the manuscript. The project was supervised by G.B.

## Competing interests
M.T. and G.B. are inventors on a granted patent about *trans*-Tango: United States Patent No.: US 10,619,155 on which Brown University is the applicant and the inventors are: Gilad Barnea, Mustafa Talay, Ethan Richman, John Szymanski, Mark Johnson, John Fisher and Nathaniel Snell. Here we report the development of *trans*-Tango MkII which is an offshoot of *trans*-Tango. Other authors declare that they have no competing interests.
