## [Peer Review File · Nature Communications]

Circuit Epistasis Analysis Reveals a Neural Pathway for Light Avoidance in *Drosophila* LarvaeEditorial Note: This manuscript has been previously reviewed at another journal that is not operating a transparent peer review scheme. This document only contains reviewer comments and rebuttal letters for versions considered at *Nature Communications*. Mentions of the other journal have been redacted.

REVIEWER COMMENTS

Reviewer #1 (Remarks to the Author):

Behavioral responses to sensory cues are manifold. A simple and often quite robust initial response is attraction or repulsion.

Drosophila larvae are typically repulsed light, a behavior that may be assessed using simple or sophisticated assays.

While the early pathways (photoreceptors and target neurons) have been identified and studied, only little is known regarding higher neural pathways. One specific aspect that has been identified is the link between Rh5-PRs and Ptth neurons.

The current manuscript aims to provide a link between the PRs and the Ptth neurons by using a modified trans-tango technique.

The approach is highly relevant and being able to link sensory neurons to neural circuits of great importance and wide interest without any doubt.

From that perspective the manuscript is timely, elegant and I feel that it will make an important contribution to the field.

My main concern lies in the format of the manuscript and presentation of the data. I can't help to feel this is a direct transfer of an initial submission to [redacted], which correspondingly is very short and, in many sections, too shallow. Since Nature Communications allows sufficient space to describe experiments in the results section as well as the corresponding figures it would be beneficial if the authors take advantage of the available format.

One particularly interesting point of the current is the adaptation of the trans-Tango system to the larva. However, the section explaining the adaptations as well as its usefulness and weakness remains rather shallow. It would be beneficial that this is better explained, both in the main text as well as in the methods. Some of the most relevant points that come to mind are:

-What is the temporal dynamic? since many Gal4 drivers in the larva show differential expression between different larval stages this is quite critical and could correspondingly also be further assessed.

- Does it work for sensory neurons as well as for CNS neurons? There is some preliminary data on this in the supplement on a OR.

I do agree that the olfactory system with its well described circuit from ORN to MB and LH would be a perfect object to proof that the system works as one would hope.

On the methodological part this section is also rather short. It would be helpful for the reader to know what adaptations have been made from the original version, why they have been implemented and how they effect the performance of the constructs. What are the strength -and potentially weaknesses- of the new version and what are aspects that other researchers may want to pay attention to?

Along the line of circuit organization: The early visual circuit has been mapped using an EM-based connectomics (in L1) as well as Gal4 drivers between L1 and L3.

From this perspective in L1 the circuit is well known for Rh5-targets as well as Rh6-targets.

With the modified trans-Tango system, the authors would have the opportunity to proof

that the genetic approach is indeed adequate to identify circuits and that there is no noise, or incorrectly labelled synaptic partners. Rh6 PRs only have 6 target neurons – 2 IOLPs and 4 Pdf-LaNs, which would easily be identifiable using confocal microscopy. Rh5 have 5 PNs and + 4 LaNs as targets, which numerically are also feasible to identify. These experiments would indeed proof the validity of the approach.

Minor points:

-For the light activation (both regular as well as red-light) I would be that also the wavelength of the light source is measured and ideally the light intensity calculated using a photospectrometer. I feel this is relevant as in the past conflicting results from different groups with no proper light-source analysis led to confusion in the field.
-Similarly, TNT has been shown to cause developmental defects, an inducible inhibition (shibireTs or similar) might resolve this issue.

Sincerely
Simon Sprecher

Reviewer #2 (Remarks to the Author):

The goal of this manuscript by Sorkac et al. is twofold: (1) to develop and present a new version of the transsynaptic tracer trans-Tango-MkII compatible with applications in the *Drosophila* larva; (2) to apply trans-Tango-MkII to map the neural circuit underlying light-avoidance behavior in the larva. While part of the larval visual circuit has been mapped through EM tracing and functional inspections, its reconstruction is incomplete and, as noted by the authors, published results can be partly conflicting. As a result, our current understanding of the organization of the larval visual system is partial. Additional work to complete the mapping of this circuit will be valuable to the community. The present manuscript contributes to these efforts.

While the present manuscript has the potential to fill both a technical and a conceptual gap in the field, it falls short of achieving any of these two objectives. First, the presentation of trans-Tango-MkII is very succinct with all technical information relegated to a supplementary figure. It is difficult for the reader to tell how the tool was adapted to the larva to reduce background noise in the central nerve cord and whether this adaption was successful (see below). It would be appropriate for the authors to benchmark trans-Tango-MkII further before applying it to clear confusion in a complex neural circuit. Second, candidate synaptic connections obtained by the authors with trans-Tango-MkII were not confirmed by another technique. Not surprisingly, the "epistatic" analysis conducted with the inward-rectifying potassium channel Kir2.1 and Chrimson yield results that do not simply fit with a simple linear pathway.

Although the manuscript reveals new players that are part of the larval visual system, it leaves the reader with a circuit where the level of confidence of multiple connections is low. For these reasons, I do not think that the authors have effectively laid down a "framework for studying more complex nervous systems and behaviors", as stated in the abstract. I have no doubt that trans-Tango-MkII is a powerful tool to advance neural-circuit tracing in general, and to complete larval visual system in particular. However, a more thorough anatomical and functional inspections will be necessary to establish a new methodology to be more largely adopted by the fly community, and beyond. The loss of function and gain of function experiments included in the manuscript represent excellent steps in this direction. Yet the analysis should be brought to more satisfying conclusions. Below is a list of suggestions to help the authors reach this goal.

Major comments/suggestions:

1. What is the principle behind the design of trans-Tango-MkII? In line 51, the authors explain that the original trans-Tango reagent had limited utility in the larva due to background noise. trans-Tango-MkII is supposed to address this limitation. How is this achieved? Can the rationale behind the improvement in design spelled out? The results of Extended Data Fig. 1C show that trans-Tango-MkII produces reduced unspecific background expression in the ventral nerve cord. However, it is surprising to find that the downstream partners of the Or42a ORN appear to cover half of the brain lobes. The EM reconstruction of the larval antennal lobe (Berck et al., eLife 2016) would not predict such a wide set of post-synaptic partners. Are all these neurons genuine postsynaptic partners? How specific is trans-Tango-MkII? It is essential to benchmark the new version of trans-Tango with a gal4 driver line specific to the olfactory projection neurons (e.g., GH146). In addition, does the new method circumvent previous weakness in signal when larvae are at 25C versus 18C?

2. Lines 77-80: Where is the specificity of the two driver lines that label the 5th-LaN neuron characterized? Could the authors explain how these two Gal4 drivers were identified? The reference cited in the main text (Jenett et al. 2012) is generic and, as far the reviewer can tell, it does not discuss the expression of specific lines used by the authors. In Figure 2a, why does the images show very limited cell projection? Since the images represent maximum projection along the z-axis, expression of the fluorescent marker in the axons should be visible in these images. Is this a limitation of sporadic expression from larval trans-Tango-MkII? This point should be discussed explicitly to interpret the images and the presence/absence of co-labeling.

3. Figure 3b: The use of trans-Tango-MkII strongly suggests that DN1s are not postsynaptic to the 5th-LaN whereas DN2s are. These results are compatible with loss-of-function experiments of Fig. 3a. Given that the specificity of trans-Tango-MkII does not seem to be perfect (see point 1, above), it would be important to validate this result with another technique to establish connectivity between pre and postsynaptic candidates. Could the authors combine optogenetic stimulation with Chrimson and functional imaging with GCaMP to show that acute activation of the 5th-LaN is sufficient to activate DN2 but not DN1? The existence of a cry-LexA should make this experiment possible for DN1. Alternatively, it would be good to corroborate the results of trans-Tango-MkII with X-RASP (or equivalent) for the connectivity of the 5th-LaN and DN1s and DN2s.

4. What is the rationale for conducting the loss-of-function experiments of Fig. 4 in the Rh5 null background given that red light is not supposed to be detected by the Bolwig's organs? The observation that the activation Pdf-LaN is sufficient to produce positive phototaxis is really interesting. The Gal80 experiments of Fig. 4e and 4f are neat. What hypothesis do the authors favor about where Pdf-LaN fits in the visual pathway of the larva? Can it be ruled out that Pdf-LaN interacts with the Rh5-PTTH pathway outlined in Fig 4g? Can the authors ascertain that Pdf-LaN is not presynaptic to PTTH (lines 156-157) in Fig 2b and 2c? It would be important establish this result further by expressing Chrimson in Pdf-LaN (with Pdf-LaN-LexA) and GCaMP in PTTH.

5. In the Extended Data Fig. 6, a loss of function screen was conducted for dim light avoidance. The DN1s+Pdf-LaN demonstrates a defect. Could the Pdf-Gal80 line be used to make the conclusions about dim-light detection specific to DN1s without the addition of the Pdf-LaN loss of function?

6. Figure 4: Although larvae are mostly blind to red light, abrupt stimulation with red light can produce a startle (aversive) response. The negative controls shown in Fig. 4 suggest that startle responses should be low in the half-plate light avoidance assay. It would nonetheless be important to demonstrate the lack of avoidance of the red side of plate in Rh5 null larvae and w1118 larvae.

Minor comments:

- **Lines 56-57: Could the authors explain why the pacemaker clock neurons in the larval visual system are attractive candidates for neurons that connect the Rh5 photoreceptors to the PTTH neurons?**
- **Panels 2b and 2c have two cell types present. It may aid in the reader accessibility of these fluorescent images to have labels of the DN1/2 versus the PDF-LaNs.**
- **Line 108: Rh5 mutants are deficient in light avoidance behavior. While it is unclear that this result is directly shown in ref. 5 (Hassan... Campos 2006), the work of Humberg ... Sprecher (Nat. Com. 208) has clearly established a loss of function in Rh5 produces a deficit in the ability of larvae to turn away from the light. This reference should be added to the manuscript.**
- **Lux to W/m^2 differs depending on the wavelength of light. The intensity used by the authors should be reported in W/m^2 for the red light.**
- **The class IV multidendritic neurons in the body walls take part to photophobic responses (Xiang...Jan, Nature 2010). Can the authors comment on whether/how this pathway is expected to interact with the PTTH pathway?**

Response to REVIEWER COMMENTS

Reviewer #1 (Remarks to the Author):

Behavioral responses to sensory cues are manifold. A simple and often quite robust initial response is attraction or repulsion.

Drosophila larvae are typically repulsed light, a behavior that may be assessed using simple or sophisticated assays.

While the early pathways (photoreceptors and target neurons) have been identified and studied, only little is known regarding higher neural pathways. One specific aspect that has been identified is the link between Rh5-PRs and Ptth neurons.

The current manuscript aims to provide a link between the PRs and the Ptth neurons by using a modified trans-tango technique.

The approach is highly relevant and being able to link sensory neurons to neural circuits of great importance and wide interest without any doubt.

From that perspective the manuscript is timely, elegant and I feel that it will make an important contribution to the field.

My main concern lies in the format of the manuscript and presentation of the data. I can't help to feel this is a direct transfer of an initial submission to [redacted], which correspondingly is very short and, in many sections, too shallow. Since Nature Communications allows sufficient space to describe experiments in the results section as well as the corresponding figures it would be beneficial if the authors take advantage of the available format.

We would like to thank Dr. Sprecher for the kind words about the elegance and timeliness of our study. Dr. Sprecher is correct that our description of the technique was brief due to the length limitations of [redacted] where our manuscript was initially submitted.

As the reviewer suggested we substantially elaborated upon our description in the results section of the revised manuscript.

One particularly interesting point of the current is the adaptation of the trans-Tango system to the larva. However, the section explaining the adaptations as well as its usefulness and weakness remains rather shallow. It would be beneficial that this is better explained, both in the main text as well as in the methods. Some of the most relevant points that come to mind are:

-What is the temporal dynamic? since many Gal4 drivers in the larva show differential expression between different larval stages this is quite critical and could correspondingly also be further assessed.

We thank the reviewer for this comment and to address it, we included a full figure characterizing the temporal dynamic properties of *trans-Tango* MkII using Rh5 and Rh6

drivers in L1, L2 and L3 (Supplementary Fig. 3). We also added a full paragraph to the text of the results section (lines 106-130).

*- Does it work for sensory neurons as well as for CNS neurons? There is some preliminary data on this in the supplement on a OR.
I do agree that the olfactory system with its well described circuit from ORN to MB and LH would be a perfect object to proof that the system works as one would hope.*

Dr. Sprecher raises an important point here. We correspondingly characterized *trans*-Tango MkII by initiating it from the GH146-Gal4 driver expressed in olfactory projection neurons and revealing their postsynaptic partners - the Kenyon cells in the mushroom bodies, as expected (Supplementary Fig. 1 f, g). We also revised the text accordingly (lines 98-104).

On the methodological part this section is also rather short. It would be helpful for the reader to know what adaptations have been made from the original version, why they have been implemented and how they effect the performance of the constructs. What are the strength -and potentially weaknesses- of the new version and what are aspects that other researchers may want to pay attention to?

We would like to thank Dr. Sprecher for this comment. This is one of the places where we kept it short because of the format requirements of the original submission. In the revised manuscript, we elaborated on how the *trans*-Tango MkII construct was devised. We also included comparison on how *trans*-Tango MkII works at 18°C and 25°C (Supplementary Fig. 1c, d). In addition, we performed *trans*-Tango MkII experiments in adults (Supplementary Fig. 2) and observed high background noise indicating that it should not be used in adults where the original *trans*-Tango yields better results. We added two paragraphs of text accordingly (lines 73-104).

Along the line of circuit organization: The early visual circuit has been mapped using an EM-based connectomics (in L1) as well as Gal4 drivers between L1 and L3.

From this perspective in L1 the circuit is well known for Rh5-targets as well as Rh6-targets.

With the modified trans-Tango system, the authors would have the opportunity to proof that the genetic approach is indeed adequate to identify circuits and that there is no noise, or incorrectly labelled synaptic partners.

Rh6 PRs only have 6 target neurons – 2 IOLPs and 4 Pdf-LaNs, which would easily be identifiable using confocal microscopy. Rh5 have 5 PNs and + 4 LaNs as targets, which numerically are also feasible to identify.

These experiments would indeed proof the validity of the approach.

We thank Dr. Sprecher for this important suggestion. Indeed, these experiments contributed significantly to our validation of *trans*-Tango MkII. We counted the

postsynaptic partners of Rh5 and Rh6 in L1 larvae and were happy to see that they correspond well to the reported numbers in the EM reconstruction (Supplementary Fig. 3). We added text to discuss this in the revised manuscript (lines 106-114).

Minor points:

-For the light activation (both regular as well as red-light) I would be that also the wavelength of the light source is measured and ideally the light intensity calculated using a photospectrometer. I feel this is relevant as in the past conflicting results from different groups with no proper light-source analysis led to confusion in the field.

We measured the light intensity and added the corresponding information to the text (lines 143, 202, 226). We also added the instruments used to measure the light intensity in the methods section (lines 348-349, 361-362).

-Similarly, TNT has been shown to cause developmental defects, an inducible inhibition (shibireTs or similar) might resolve this issue.

We did not use TNT, but rather Kir2.1, as has been used extensively by the field in elegant studies to manipulate the activity of neurons in this circuit (e.g., Keene *et al.* 2011, Humberg and Sprecher 2018, Schlichting *et al.* 2019).

*Sincerely
Simon Sprecher*

Reviewer #2 (Remarks to the Author):

The goal of this manuscript by Sorkac et al. is twofold: (1) to develop and present a new version of the transsynaptic tracer trans-Tango-MkII compatible with applications in the Drosophila larva; (2) to apply trans-Tango-MkII to map the neural circuit underlying light-avoidance behavior in the larva. While part of the larval visual circuit has been mapped through EM tracing and functional inspections, its reconstruction is incomplete and, as noted by the authors, published results can be partly conflicting. As a result, our current understanding of the organization of the larval visual system is partial. Additional work to complete the mapping of this circuit will be valuable to the community. The present manuscript contributes to these efforts.

While the present manuscript has the potential to fill both a technical and a conceptual gap in the field, it falls short of achieving any of these two objectives. First, the presentation of trans-Tango-MkII is very succinct with all technical information relegated to a supplementary figure. It is difficult for the reader to tell how the tool was adapted to the larva to reduce background noise in the central nerve cord and whether this adaptation was successful (see below). It would be appropriate for the authors to benchmark trans-Tango-MkII further before applying it to clear confusion in a complex

neural circuit. Second, candidate synaptic connections obtained by the authors with trans-Tango-MkII were not confirmed by another technique. Not surprisingly, the "epistatic" analysis conducted with the inward-rectifying potassium channel Kir2.1 and Chrimson yield results that do not simply fit with a simple linear pathway. Although the manuscript reveals new players that are part of the larval visual system, it leaves the reader with a circuit where the level of confidence of multiple connections is low. For these reasons, I do not think that the authors have effectively laid down a "framework for studying more complex nervous systems and behaviors", as stated in the abstract. I have no doubt that trans-Tango-MkII is a powerful tool to advance neural-circuit tracing in general, and to complete larval visual system in particular. However, a more thorough anatomical and functional inspections will be necessary to establish a new methodology to be more largely adopted by the fly community, and beyond. The loss of function and gain of function experiments included in the manuscript represent excellent steps in this direction. Yet the analysis should be brought to more satisfying conclusions. Below is a list of suggestions to help the authors reach this goal.

We would like to thank the reviewer for acknowledging the contribution of our study to the field. As the reviewer suggested, we expanded on the characterization of *trans-Tango MkII* in figures (Supplementary Fig. 1,2,3) and in text (lines 73-130). For instance, we benchmarked *trans-Tango MkII* by comparing its results to the EM reconstruction of the larval visual system. We counted the postsynaptic partners of Rh5 and Rh6 in L1 larvae as revealed by *trans-Tango MkII* and showed that they correspond well to the reported numbers in the EM reconstruction. Finally, we removed the word linear from the title and the new title is as follows:

“Circuit Epistasis Analysis Reveals a Neural Pathway for Light Avoidance in *Drosophila* Larvae”

Major comments/suggestions:

1. What is the principle behind the design of trans-Tango-MkII? In line 51, the authors explain that the original trans-Tango reagent had limited utility in the larva due to background noise. trans-Tango-MkII is supposed to address this limitation. How is this achieved? Can the rationale behind the improvement in design spelled out? The results of Extended Data Fig. 1C show that trans-Tango-MkII produces reduced unspecific background expression in the ventral nerve cord. However, it is surprising to find that the downstream partners of the Or42a ORN appear to cover half of the brain lobes. The EM reconstruction of the larval antennal lobe (Berck et al., eLife 2016) would not predict such a wide set of post-synaptic partners. Are all these neurons genuine postsynaptic partners? How specific is trans-Tango-MkII? It is essential to benchmark the new version of trans-Tango with a gal4 driver line specific to the olfactory projection neurons (e.g., GH146). In addition, does the new method circumvent previous weakness in signal when larvae are at 25C versus 18C?

We would like to thank the reviewer for this comment. In order to address these questions, we replaced Supplementary Fig. 1 in the revised manuscript. The comment about the postsynaptic partners of Or42a ORNs stems from the overexposure of the HA-staining in the original version of the figure. In the revised manuscript, we replaced the panels with better taken pictures to better reflect the results of *trans*-Tango MkII. We are also discussing the specificity and the false positives of *trans*-Tango MkII by comparing the number of projection neurons we observe to the results of the EM reconstruction. In addition, as the reviewer suggested, we initiated *trans*-Tango MkII from GH146-expressing neurons and revealed the Kenyon cells of the mushroom bodies as postsynaptic partners, as expected. In panels c and d, we addressed the reviewer's question about temperature. We also revised the text accordingly by adding two paragraphs in the revised manuscript (lines 73-104).

2. Lines 77-80: Where is the specificity of the two driver lines that label the 5th-LaN neuron characterized? Could the authors explain how these two Gal4 drivers were identified? The reference cited in the main text (Jenett et al. 2012) is generic and, as far the reviewer can tell, it does not discuss the expression of specific lines used by the authors.

The reviewer raises an important point about the two driver lines. In the revised manuscript, we added the corresponding reference that characterized the line R19C05-Gal4. In addition, we added Supplementary Fig. 6 in which we characterized the other line, R54D11-Gal4. Further, we added a panel where the whole CNS of the larva is shown along with another panel where we co-stain for PER and PDF to demonstrate that the line is expressed in the 5th-LaN.

In Figure 2a, why does the images show very limited cell projection? Since the images represent maximum projection along the z-axis, expression of the fluorescent marker in the axons should be visible in these images. Is this a limitation of sporadic expression from larval trans-Tango-MkII? This point should be discussed explicitly to interpret the images and the presence/absence of co-labeling.

This problem stems from the Gal4 line (R54D11) we are using to access the 5th-LaN. Unfortunately, the driver is very weak and GFP expression in neuronal processes is observed in less than 10% of the brains using our reporter. Using a stronger reporter might reveal the processes more effectively. We would like to draw the reviewer's attention to Supplementary Fig. 6a where the processes of the 5th-LaNs are visible albeit very dimly. By contrast, in Supplementary Fig. 6b we do not see the processes of the 5th-LaN at all.

3. Figure 3b: The use of trans-Tango-MkII strongly suggests that DN1s are not postsynaptic to the 5th-LaN whereas DN2s are. These results are compatible with loss-of-function experiments of Fig. 3a. Given that the specificity of trans-Tango-MkII does

not seem to be perfect (see point 1, above), it would be important to validate this result with another technique to establish connectivity between pre and postsynaptic candidates. Could the authors combine optogenetic stimulation with Chrimson and functional imaging with GCaMP to show that acute activation of the 5th-LaN is sufficient to activate DN2 but not DN1? The existence of a cry-LexA should make this experiment possible for DN1. Alternatively, it would be good to corroborate the results of trans-Tango-MkII with X-RASP (or equivalent) for the connectivity of the 5th-LaN and DN1s and DN2s.

As we answered to the reviewer's comment 1, the benchmarking of *trans*-Tango MkII by comparing it to the EM reconstruction, in fact, indicates that *trans*-Tango MkII reveals connections that are very similar.

While using calcium imaging could be a good idea, it is not optimal for our purposes because it would not necessarily reveal only monosynaptic connections. By contrast, the use of a synaptic version of GRASP is a good idea to confirm *trans*-Tango MkII results since it would reveal direct synaptic connections. Indeed, we use t-GRASP to answer the reviewer's next comment.

However, we could not perform the GRASP experiments for DN2s since a LexA line that labels these neurons is not available. Although there is a LexA version of R54D11-Gal4, it does not drive expression in the 5th-LaN preventing us from using it in GRASP experiments. For DN1s, the LexA driver also expresses in Pdf-LaNs. Therefore, it would not be possible to discern whether the GRASP signal reflects the connection between the LaNs or a connection, not predicted by *trans*-Tango MkII, between the 5th-LaN and DN1s.

4. What is the rationale for conducting the loss-of-function experiments of Fig. 4 in the Rh5 null background given that red light is not supposed to be detected by the Bolwig's organs?

We wanted to avoid any other behavioral response that might be caused by the red light as the reviewer pointed out in comment 6. However, since we performed the experiment that the reviewer suggested in this comment (*w*¹¹¹⁸ and Rh5 mutants under red light), we now know that the red light does not cause any photophobic behavior. In addition, we wanted to rescue the Rh5 defects using circuit epistasis, an idea we borrowed from genetics, and for that we had to use Rh5 mutant animals.

The observation that the activation Pdf-LaN is sufficient to produce positive phototaxis is really interesting. The Gal80 experiments of Fig. 4e and 4f are neat. What hypothesis do the authors favor about where Pdf-LaN fits in the visual pathway of the larva? Can it be ruled out that Pdf-LaN interacts with the Rh5-PTTH pathway outlined in Fig 4g? Can the authors ascertain that Pdf-LaN is not presynaptic to PTTH (lines 156-157) in Fig 2b

and 2c? It would be important establish this result further by expressing Chrimson in Pdf-LaN (with Pdf-LaN-Lexa) and GCaMP in PTTH.

We would like to thank the reviewer for their kind words. We cannot rule out that Pdf-LaNs interact with this pathway, especially since ablating Pdf-LaNs increases the GCaMP signal in PTTH neurons as shown by Gong *et al.* in 2010. The same study argued, based on GRASP, that Pdf-LaNs were presynaptic to PTTH neurons whereas our *trans*-Tango MkII results indicated no such connection. We would like to point out that the version of GRASP used in this study relies on the reconstitution of GFP domains that are fused to CD4, a protein that does not exclusively localize to the synapse. In the revised version of the manuscript, we wanted to confirm these results by using a synaptic version of GRASP, t-GRASP, and we do not observe reconstituted GFP signal (Supplementary Fig. 5), corroborating our *trans*-Tango MkII results. We added extensive discussion of this point in the revised manuscript (lines 155-160, 250-255)

5. In the Extended Data Fig. 6, a loss of function screen was conducted for dim light avoidance. The DN1s+Pdf-LaN demonstrates a defect. Could the Pdf-Gal80 line be used to make the conclusions about dim-light detection specific to DN1s without the addition of the Pdf-LaN loss of function?

We would like to thank the reviewer for this suggestion. We conducted the experiments that the reviewer asked for (Supplementary Fig. 11b). When we silenced only DN1 neurons via Pdf-Gal80, we observed that there was a significant difference compared to Gal4 only controls. However, this significance was not observed when compared to Kir2.1+Gal80 controls although the p value was very low ($p=0.0674$). Since silencing DN1s or DN1s+Pdf-LaNs does not result in significantly different performance indices, we think that it is safe to say that DN1s are playing an important role in dim light avoidance. In addition, we do not observe the inhibitory effect of Pdf-LaNs in dim light conditions, indicating that Pdf-LaNs are not active in response to dim light.

6. Figure 4: Although larvae are mostly blind to red light, abrupt stimulation with red light can produce a startle (aversive) response. The negative controls shown in Fig. 4 suggest that startle responses should be low in the half-plate light avoidance assay. It would nonetheless be important to demonstrate the lack of avoidance of the red side of plate in Rh5 null larvae and w1118 larvae.

We performed the experiment that the reviewer suggested. Neither Rh5 null mutants, nor w^{1118} larvae, exhibited avoidance of red light (Supplementary Fig. 9).

Minor comments:

- Lines 56-57: Could the authors explain why the pacemaker clock neurons in the larval

visual system are attractive candidates for neurons that connect the Rh5 photoreceptors to the PTTH neurons?

We added the following phrase (highlighted in yellow) and reference to the sentence:

“The pacemaker clock neurons in the larval visual system are attractive candidates since they were previously implicated in light avoidance¹.”

- Panels 2b and 2c have two cell types present. It may aid in the reader accessibility of these fluorescent images to have labels of the DN1/2 versus the PDF-LaNs.

We added the labels on the figures.

- Line 108: Rh5 mutants are deficient in light avoidance behavior. While it is unclear that this result is directly shown in ref. 5 (Hassan... Campos 2006), the work of Humberg ... Sprecher (Nat. Com. 208) has clearly established a loss of function in Rh5 produces a deficit in the ability of larvae to turn away from the light. This reference should be added to the manuscript.

We thank the reviewer for this comment and we have added the corresponding reference.

- Lux to W/m² differs depending on the wavelength of light. The intensity used by the authors should be reported in W/m² for the red light.

We measured the light intensity and added the corresponding information to the text (lines 143, 202, 226). We also added the instruments used to measure the light intensity in the methods section (lines 348-349, 361-362).

- The class IV multidendritic neurons in the body walls take part to photophobic responses (Xiang...Jan, Nature 2010). Can the authors comment on whether/how this pathway is expected to interact with the PTTH pathway?

At this point we cannot make any comments about the interactions of these two pathways, we added this to the text:

“It is noteworthy that a third, independent system has been reported in which a gustatory receptor mediates photophobic response to high-intensity light in class IV multidendritic neurons²³. We do not have any information as to at which level these pathways might meet, if at all, before the motor neurons.”

REVIEWER COMMENTS

Reviewer #1 (Remarks to the Author):

The authors have addressed all points in detail. I particularly like that the revised manuscript is less dense, it reads very well and follows a clear logic. It is also really good to see that the tool is indeed widely usable with the example included of the olfactory circuit.

It is a powerful technique and the paper nicely shows how it can be used to functionally and genetically dissect described circuits from EM-connectomics. I strongly feel it's an impacting piece for the field.

Simon Sprecher

Reviewer #2 (Remarks to the Author):

An initial concern surrounded the scarce description of the new tool, trans-Tango MkII. In their revised manuscript, the authors added a description of trans-Tango MkII and the rationale behind the changes made to the original version of trans-Tango. The authors also addressed several technical concerns in supplementary material. They documented the expression pattern of the Gal4 line (R54D11) used in the analysis of the larval visual pathway. They included t-GRASP data that reinforces a key conclusion reached with trans-Tango MkII (PTTH neurons are not postsynaptic to Pdf-LaNs). This result demonstrates that a generic GRASP might have unspecific binding, which misled an earlier understanding of the Pdf-LaNs connectivity. New loss of function experiments corroborate results related to the visual pathway reconstruction. Finally, the authors provided additional details about the conditions that improve reproducibility, including a recommended temperature of 25C and a word of caution on developmental stage. All these efforts strengthened the manuscript, which is commendable. I do not have any additional request associated with the analysis of the light avoidance pathway.

On line 60, the authors stated that "Since trans-Tango MkII fills a gap in larval circuit tracing, our approach constitutes a general framework for studying neural circuits in *Drosophila* larvae." This statement relies on the assumption that the technique is reasonably specific, that it does not lead to a large number of false positives. In my view, the data provided in the Supplementary Figure 1 are insufficient to establish the validity of this assumption. Instead, they suggest that trans-Tango MkII is relatively leaky. The leakiness of trans-Tango MkII does not necessarily imply that the tool is bad. The authors have clearly shown that trans-Tango MkII can assist the mapping of a neural circuit in combination with other tools. While no tool is perfect and imperfect tools improve over time, it is important that the authors discuss the caveats of their technique. This problem should be addressed openly through an assessment of the post-synaptic candidates found in Supplementary Figure 1 — a figure that is supposed to benchmark the performances of trans-Tango MkII. A quantitative comparison should be made with EM connectivity data (see below). Based on these results, the specificity of trans-Tango MkII should be candidly discussed. Even if the tool's specificity turns out to be modest, it would still be valuable for screening or confirmation purposes.

1. In Supplementary Figure 1a-d, Or42a OSN is used as the pre-synaptic site. Using the olfactory system of the larva to benchmark the performances of trans-Tango MkII is sensible given the existence of a complete EM reconstruction of the larval antennal lobe. The authors added a valuable discussion of results pertaining to the labeling of the olfactory projection neurons. However, it is obvious that a large number of post-synaptic candidates are distinct from olfactory projection neurons and fall outside the antennal lobe. These candidates appear to be in the SEZ region. Does the EM connectivity predict such a large number of post-synaptic partners of the Or42a OSN in this region of the brain?

2. I appreciate that the authors have added new data related to the GH146-Gal4 line. Overall, the labeling of the Kenyon cells corroborates the idea that the technique labels a set of expected post-synaptic partners. However, concerns related to the specificity of the labeling persist in Supplementary Figure 1f. In addition to a close-up view on the mushroom body calyx, an image of the brain lobes and the ventral nerve cord should be shown. Are there other neurons labeled besides the Kenyon Cells? If so, how many additional cells are labeled?

Points 1 and 2 ought to be addressed quantitatively because they touch on a fundamental aspect of the technique: its specificity. Since that the authors propose an improved version of an existing technique, the improvement must be properly documented.

Response to Reviewer Comments

Reviewer #1 (Remarks to the Author):

The authors have addressed all points in detail. I particularly like that the revised manuscript is less dense, it reads very well and follows a clear logic. It is also really good to see that the tool is indeed widely usable with the example included of the olfactory circuit.

It is a powerful technique and the paper nicely shows how it can be used to functionally and genetically dissect described circuits from EM-connectomics. I strongly feel its an impacting piece for the field.

Simon Sprecher

We would like to thank Dr. Sprecher for the kind words about the manuscript.

Reviewer #2 (Remarks to the Author):

An initial concern surrounded the scarce description of the new tool, trans-Tango MkII. In their revised manuscript, the authors added a description of trans-Tango MkII and the rationale behind the changes made to the original version of trans-Tango. The authors also addressed several technical concerns in supplementary material. They documented the expression pattern the Gal4 line (R54D11) used in the analysis of the larval visual pathway. They included t-GRASP data that reinforces a key conclusion reached with trans-Tango MkII (PTTH neurons are not postsynaptic to Pdf-LaNs). This result demonstrates that a generic GRASP might have unspecific binding, which misled an earlier understanding of the Pdf-LaNs connectivity. New loss of function experiments corroborate results related to the visual pathway reconstruction. Finally, the authors provided additional details about the conditions that improves reproducibility, including a recommended temperature of 25C and a word of caution on developmental stage. All these efforts strengthened the manuscript, which is commendable. I do not have any additional request associated with the analysis of the light avoidance pathway.

We would like to thank the reviewer for these comments. We agree with the reviewer that their suggestions/comments did strengthen our manuscript.

On line 60, the authors stated that "Since trans-Tango MkII fills a gap in larval circuit tracing, our approach constitutes a general framework for studying neural circuits in Drosophila larvae." This statement relies on the assumption that the technique is reasonable specific, that it does not lead to a large number of false positives. In my view, the data provided in the Supplementary Figure 1 are insufficient to establish the validity of this assumption. Instead, they suggest that trans-Tango MkII is relatively leaky. The leakiness of trans-Tango MkII does not necessarily imply that the tool is bad. The authors have clearly shown that trans-Tango MkII can assist the mapping of a

neural circuit in combination with other tools. While no tool is perfect and imperfect tools improve over time, it is important that the authors discuss the caveats of their technique. This problem should be addressed openly through an assessment of the post-synaptic candidates found in Supplementary Figure 1 — a figure that is supposed to benchmark the performances of trans-Tango MkII. A quantitative comparison should be made with EM connectivity data (see below). Based on these results, the specificity of trans-Tango MkII should be candidly discussed. Even if the tool's specificity turns out to be modest, it would still be valuable for screening or confirmation purposes.

We would like to thank the reviewer for the suggestions to improve the assessment of *trans-Tango MkII*. Our responses to the two points raised by the reviewer are below.

1. In Supplementary Figure 1a-d, Or42a OSN is used as the pre-synaptic site. Using the olfactory system of the larva to benchmark the performances of trans-Tango MkII is sensible given the existence of a complete EM reconstruction of the larval antennal lobe. The authors added a valuable discussion of results pertaining to the labeling of the olfactory projection neurons. However, it is obvious that a large number of post-synaptic candidates are distinct from olfactory projection neurons and fall outside the antennal lobe. These candidates appear to be in the SEZ region. Does the EM connectivity predict such a large number of post-synaptic partners of the Or42a OSN in this region of the brain?

In response to the reviewer's suggestions, we counted the number of postsynaptic neurons labeled when *trans-Tango MkII* is initiated from Or42a-expressing OSNs. In five brains we counted an average of 22 neurons per side of the brain. The EM reconstruction revealed 14 and 16 neurons on each side of the brain to be postsynaptic to Or42a OSNs (16 and 20 if single synapse connections are counted). Hence, we conclude that *trans-Tango MkII* labels more neurons than the EM reconstruction. We discussed this in the text mentioning that these could be true false positives or that this increase might be due to connectivity changes from first instar larva (used for the EM reconstruction) to third instar larva (used in the *trans-Tango MkII* analysis). We cannot conclude either way.

That said, we agree with the reviewer that the signal in the SEZ is more prominent than expected. Based on the EM reconstruction, at least two of the postsynaptic partners (Keystone and IAL-1 neurons) have processes in the SEZ. However, the *trans-Tango MkII* postsynaptic signal in the SEZ seems to be denser than just the processes of these neurons. We discussed this in the revised text as well.

Accordingly, we added panel e to Supplementary Figure 1 showing a close-up view of the postsynaptic partners revealed in panel d. We also added lines 93-104 and 137-138 to discuss our results in the main text.

2. I appreciate that the authors have added new data related to the GH146-Gal4 line. Overall, the labeling of the Kenyon cells corroborates the idea that the technique labels a set of expected post-synaptic partners. However, concerns related to the specificity of the labeling persist in Supplementary Figure 1f. In addition to a close-up view on the mushroom body calyx, an image of the brain lobes and the ventral nerve cord should be shown. Are there other neurons labeled besides the Kenyon Cells? If so, how many additional cells are labeled?

The reviewer raises an important point here. The reason we only added a close-up view of the mushroom body calyx is that the GH146-Gal4 line used in this experiment is not specific to olfactory projection neurons. Hence, the *trans-Tango MkII* postsynaptic signal is present in many places outside the olfactory circuit.

As was previously reported (Moraru, Egger, Bao, Sprecher; 2012 and Wang, Haenfler, Leel 2011), we observe GH146-Gal4 expression in the ventral nerve cord and in the optic neuroepithelium. The expression pattern of the Gal4 line hinders our ability to perform the analysis requested by the reviewer. The image below shows the expression of GH146-Gal4 as revealed by UAS-GFP (Green: GFP, Blue: neuropil). Please note the extensive expression in the optic lobes (arrows) and along the ventral nerve cord (bracket).

Points 1 and 2 ought to be addressed quantitatively because they touch on a fundamental aspect of the technique: its specificity. Since that the authors propose an improved version of an existing technique, the improvement must be properly documented.

As we discussed above, we performed a quantitative analysis of the postsynaptic partners of Or42a-expressing ORNs and compared it to the EM reconstruction of the larval olfactory system in response to the reviewer's request. We cannot perform the same analysis for the olfactory projection neurons since the GH146 driver is not specific to these neurons.

Reviewers' comments:

Reviewer #2 (Remarks to the Author):

I thank the authors for the extra work they have done to discuss the specificity of their new tool, trans-Tango MkII. The quantitative inspection of the number of post-synaptic candidates downstream from the Or42a OSN is helpful. The close-up view of Supplementary Figure 1e conveys the idea that the number of post-synaptic candidates is compatible with the statistics added in the main text. It is in relatively good agreement with the EM connectivity. Is the additional material sufficient to mitigate a major concern about the specificity of the technique? The answer is yes - from what I can tell trans-Tango MkII does not label a very large number of neurons downstream from the Or42a OSN, which was not obvious in earlier versions of the manuscript. Moreover, the authors disclose the limits of their tool in the discussion. This revision addresses my concerns.

Do I find the the extra material satisfying to document the technique's specificity? The answer is no. The revision is bare-bones. In addition, the data related to GH146-Gal4 are troublesome. I was not aware of the extensive expression of this driver line in the optic lobe and the ventral nerve cord. If the authors knew about this issue, why did they include data related to the GH146 driver line without discussing the limitations of this control? The purpose of the control was to benchmark the specificity of trans-Tango MkII through its application with two narrowly-expressed Gal4 lines. In retrospect, GH146-Gal4 was not a good choice to test specificity. While I don't hold the authors responsible for the broad expression of Gh146-Gal4, I am surprised they included the control anyway in their Supplementary Figure 1. The idea was not to please an overzealous reviewer but to improve the quality of the paper and its contribution to the field.

Reviewers' comments:

Reviewer #2 (Remarks to the Author):

I thank the authors for the extra work they have done to discuss the specificity of their new tool, trans-Tango MkII. The quantitative inspection of the number of post-synaptic candidates downstream from the Or42a OSN is helpful. The close-up view of Supplementary Figure 1e conveys the idea that the number of post-synaptic candidates is compatible with the statistics added in the main text. It is in relatively good agreement with the EM connectivity. Is the additional material sufficient to mitigate a major concern about the specificity of the technique? The answer is yes - from what I can tell trans-Tango MkII does not label a very large number of neurons downstream from the Or42a OSN, which was not obvious in earlier versions of the manuscript. Moreover, the authors disclose the limits of their tool in the discussion. This revision addresses my concerns.

We would like to thank the reviewer for the kind words about our revisions.

Do I find the the extra material satisfying to document the technique's specificity? The answer is no. The revision is bare-bones. In addition, the data related to GH146-Gal4 are troublesome. I was not aware of the extensive expression of this driver line in the optic lobe and the ventral nerve cord. If the authors knew about this issue, why did they include data related to the GH146 driver line without discussing the limitations of this control? The purpose of the control was to benchmark the specificity of trans-Tango MkII through its application with two narrowly-expressed Gal4 lines. In retrospect, GH146-Gal4 was not a good choice to test specificity. While I don't hold the authors responsible for the broad expression of Gh146-Gal4, I am surprised they included the control anyway in their Supplementary Figure 1. The idea was not to please an overzealous reviewer but to improve the quality of the paper and its contribution to the field.

We are shocked by the reviewer's comments because we feel that we should not be penalized for performing experiments that the reviewer had requested.

First of all, we do not think that the revision is bare-bones. As the reviewer requested, we performed the quantitative analysis of the postsynaptic partners of Or42a-expressing olfactory receptor neurons. As the reviewer commented, our results are "in relatively good agreement with the EM connectivity". We would also like to draw attention to the fact that we perform the *trans*-Tango MkII analysis in third instar larvae whereas the EM reconstruction is in first instar larvae. This could potentially explain the changes in connectivity in the olfactory circuit. We also discuss this part in the main text:

"The fact that we see more neurons via trans-Tango MkII might reveal changes in the connections between first and third instar larvae. However, although some of the neurons identified by the EM reconstruction have processes in the suboesophageal

zone, the density of the innervation in this area might mean that *trans-Tango MkII* exhibits some false positive signal. Nonetheless, *trans-Tango MkII* reveals the expected connections in this circuit.”

Furthermore, we performed the same quantitative analysis in the visual system by initiating *trans-Tango MkII* from two different subsets of neurons using two distinct Gal4 drivers (Rh5 and Rh6). We compared the results we obtain from *trans-Tango MkII* with the EM reconstruction of the larval visual system. From our analysis, we concluded that the number of neurons revealed by *trans-Tango MkII* is in accordance with the EM data in the visual system. The reviewer wanted us to benchmark *trans-Tango MkII* using two narrowly-expressed drivers. We, therefore, feel that validating the technique using three narrowly-expressed drivers in different sensory systems should be sufficient.

Second, the use of GH146-Gal4 was suggested by the reviewer themselves for initiating *trans-Tango MkII* from olfactory projection neurons. Since the expression pattern of this driver line was documented a decade ago, we assumed that the reviewer was aware of the ectopic expression of the driver in tissues outside of the olfactory circuit. This precludes the use of this line in performing a quantitative assessment of the *trans-Tango MkII* system. Accordingly, we did not perform this analysis. We used this driver in order to assess whether *trans-Tango MkII* can be used in revealing connections within the central nervous system and our experiments revealed that it can. We were extremely prudent in order not to overinterpret the results. Indeed, our conclusions from the experiment where we used GH146-Gal4 are limited and very carefully worded. Nevertheless, to avoid any confusion that the readers might have, we altered the text to mention the ectopic expression of the GH146-Gal4 driver and added the relevant references. The text now reads:

“Having successfully used *trans-Tango MkII* to trace connections from the periphery to the CNS, we next wished to implement it to reveal connections within the CNS. One such easily identifiable connection exists in the mushroom body calyx between the PNs and the Kenyon Cells^{9,10}. To access a subset of PNs, we employed the commonly used GH146 driver that also expresses in cells outside the olfactory circuit^{11,12,13}. When we initiated *trans-Tango MkII* from GH146-expressing PNs we observed postsynaptic signal in Kenyon cells as expected (Supplementary Fig. 1g, h).”

REVIEWERS' COMMENTS

Reviewer #1 (Remarks to the Author):

The authors substantially extended the analysis of the larval trans-Tango system. They went beyond the initial scope of using it for the larval visual system and show that the tool is of great use for other circuits. The core points raised by another review are more related to how clean a specific line is- we all know that some lines are more leaky than others and even in the most elegant intersection approaches the responders may matter (there are a few beautiful examples from colleagues in HHMI Janelia who systematically tested 10x vs 20x UAS lines). The overall data here points that the system works very well and I therefore recommend publication of the manuscript without any further edits.

Reviewer #1 (Remarks to the Author):

The authors substantially extended the analysis of the larval trans-Tango system. They went beyond the initial scope of using it for the larval visual system and show that the tool is of great use for other circuits. The core points raised by another review are more related to how clean a specific line is- we all know that some lines are more leaky than others and even in the most elegant intersection approaches the responders may matter (there are a few beautiful examples from colleagues in HHMI Janelia who systematically tested 10x vs 20x UAS lines). The overall data here points that the system works very well and I therefore recommend publication of the manuscript without any further edits.

We would like to thank the reviewer for the kind comments about the system and the manuscript.